# Study of Viral Coinfection of the *Ixodes persulcatus* Ticks Feeding on Humans in a Natural Focus of the South of the Far East

**DOI:** 10.3390/microorganisms11071791

**Published:** 2023-07-12

**Authors:** Galina N. Leonova, Larisa M. Somova, Svetlana A. Abramova, Evgeniy V. Pustovalov

**Affiliations:** 1G.P. Somov Research Institute of Epidemiology and Microbiology, Rospotrebnadzor, 690087 Vladivostok, Russia; l_somova@mail.ru (L.M.S.); svetochey99@mail.ru (S.A.A.); pustovalov.ev@dvfu.ru (E.V.P.); 2Department of Information and Computer Systems, Far Eastern Federal University, 690922 Vladivostok, Russia

**Keywords:** coinfection, tick *Ixodes persulcatus*, virus isolate, tick-borne encephalitis virus (TBEV), ectromelia virus, biological properties of co-isolate, electron microscopy, south of the Russian Far East

## Abstract

The phenomenon of pathogen co-infection detected in a half-fed *Ixodes persulcatus* tick taken from a human in the south of the Far East was studied. Research was carried out on *PEK*, *Vero*, and *Vero-E6* cell lines, outbred mice, and chicken embryos using ELISA, PCR, IMFA, plaque formation, and electron microscopy. The tick contained an antigen and a genetic marker of the tick-borne encephalitis virus (TBEV). The patient had post-vaccination antibodies in a titer of 1:200, as a result of which, obviously, an antibody-dependent elimination of TBEV occurred. The tick-borne co-isolate also contained an unknown pathogen (*Kiparis-144* virus), which, in our opinion, was a trigger for the activation of chronic infection in suckling white mice. In the laboratory co-isolate, ectromelia virus was present, as evidenced by paw edema during the intradermal infection of mice, characteristic rashes on the chorioallantoic envelope of chicken embryos, and typical plaques on Vero-E6. The *Kiparis-144* virus was not pathogenic for white mice and chicken embryos, but it successfully multiplied in the *PEK*, *Vero*, and *Vero-E6* lines. Viral co-infection was confirmed by electron microscopy. Passaging on mice contributed to an increase in the virulence of the co-isolate, whose titer increased by 10,000 times by the fifth passage, which poses an epidemiological danger.

## 1. Introduction

In recent years, in the study of the tick virome, modern research approaches (metagenomic and metatranscriptomic) have been used that have revealed an enormous number of putative new pathogens [1]. Such information greatly enriches knowledge of the natural diversity of possible pathogens in different tick species [2,3]. To be aware of potentially dangerous pathogens capable of causing infection in humans and animals, it is necessary to have isolates of pathogens with disclosed molecular structure sequences and studied biological characteristics [4].

Thus, since 2009, severe fever with thrombocytopenia syndrome (SFTS) began to be registered in six provinces of China, the cause of which was not known [5,6,7,8,9]. Genome sequencing of the isolated pathogen has established that the SFTS virus belongs to a new (third) group of the *Phlebovirus* genus of the *Bunyaviridae* family. This virus has aroused great interest among researchers. It turned out that the SFTS virus is found not only in China but also in the territories adjacent to the Far East—in the Republic of Korea and Japan. In addition, its wide distribution has become known in many countries of the world [2,6,10,11].

Earlier, in 1971, the *Khasan* virus *(KHAV*) was isolated from ticks (*Haemophysalis longicornis* Neumann, 1901) collected from spotted deer (*Cervus nippon* Temmink, 1838) in the territory of the Khasansky region (south of the Primorsky Territory of the Far East). Based on biological properties, morphology [12,13], and molecular genetic characteristics, the KHAV virus was assigned to the genus *Phlebovirus* of the *Bunyaviridae* family [14]. In this regard, our attention was directed to the identification of similar and other new pathogens in natural foci in the south of the Far East.

In the last decade, great attention has been drawn to reports on the isolation of especially dangerous pathogens of viral infections. Numerous publications are known that are about new variants of the smallpox virus with non-traditional sources of isolation [15,16,17]. This applies to recently discovered pathogens of the smallpox virus, which were isolated from relatively poorly studied hosts (fish, bats, porcupines, mosquitoes, birds, and aquatic mammals) or from humans [18,19]. Noteworthy is the history of one strain of ectromelia virus isolation during intracerebral infection of laboratory white mice with homogenates of ixodid ticks [20]. Initially, the authors identified the co-infection of two viruses (mouse ectromelia virus and lactate dehydrogenase-elevating virus (LDV) belonging to the family *Arteriviridae*) in ticks, but they failed to pass LDV. Therefore, the authors of the article characterized only a new strain of ectromelia virus, ECTV (ECTV-WH) [20].

A similar situation arose in our study of a viral isolate from the tick *Ixoodes persulcatus*. During the isolation of the putative virus from a half-fed *I. persulcatus* tick taken from a person in 2016, it was possible to identify the manifestation of co-infection of viral pathogens. It was necessary to verify it and to give a comprehensive biological characterization of viruses.

## 2. Materials and Methods

### 2.1. Virus

We isolated an unknown pathogen from the *I. persulcatus* tick taken on the 5th day of bloodsucking on a 75-year-old patient. The tick bite occurred on 21 May 2016 in the territory of the natural focus in the Nadezhdinsky district (south of the Russian Far East).

An unknown-to-us pathogen was isolated from a half-fed *I. persulcatus* tick taken from a patient on 21 May 2016 in the south of the Russian Far East (Nadezhda region of Primorsky Krai). In case of primary simultaneous infection in the brain and subcutaneously of 2-day-old outbred white mice of the same litter with a 10% suspension of ticks, on the 6th day, one animal with an unclear clinic of the disease was euthanized, from which the brain was taken for further virological examination. The virus isolates from the 2nd to the 5th passages were used in the work.

### 2.2. Enzyme Immunoassay (ELISA)

The detection of the antigen of the tick-borne encephalitis virus (TBEV) in the tick homogenate was carried out by ELISA using the “VectoTBEV-antigen” kit (ZAO “Vector-Best”, Novosibirsk, Russia), according to the instructions of the test system manufacturer.

### 2.3. Real-Time PCR (qPCR)

The tick was examined for the presence of genetic markers, such as tick-borne encephalitis virus (TBE), *Borrelia burgdorferi sensu lato*, *Anaplasma phagocytophilum*, and *Ehrlichia muris/Ehrlichia chaffeensis*, by real-time polymerase chain reaction (RT-PCR) using the kit “AmpliSense TBEV, *B. burgdorferi s.l.*, *A. phagocytophilum*, *E. chaffeensis/E. muris-FL*” (Central Research Institute of Epidemiology, Moscow, Russia), according to the manufacturer’s instructions on a cycler with fluorescent detection “ROTOR-GENE Q” (QIAGEN, Hilden, Germany).

### 2.4. Indirect Immunofluorescent Antibody (IFA) Method

The indirect MFA method was used to detect the antigen in Pig embryo kidney *(PEK)*, *Vero*, and *Vero-E6* culture cells infected with test samples in 3 tubes. Cells from these tubes were collected at certain times of the experiments, and then slides were prepared from the mixture of cells on objective glasses. The antigen of the virus was detected in the cells by applying specific immune serum to the slides and, subsequently, fluorescent immunoglobulins (FITS) in the working dilution specified in the manufacturer’s instructions (Branch “MEDGAMAL”, N.F. Gamaleya NIIEM). Slides were viewed in 3 fields on a fluorescent microscope (MC-200 TF A-1120 Vienna, Austria). The average percentage of cells with the content of the fluorescent virus antigen in relation to the total number of cells in the field of view was considered.

### 2.5. Plaque Method

The plaque method was used to study the biological characterization of the virus in the *PEK*, *Vero*, and *Vero-E6* cell lines. The investigation was carried out on 24-well plates using a coating of carboxymethyl cellulose (“Signa Aldrich”, Saint Louis, MO, USA). The titer of virus samples was calculated in plaque-forming units (PFU).

### 2.6. Methods for Studying the Biological Properties of a Viral Isolate

#### 2.6.1. Virulence

The studies were carried out on models of 2-day-old and 4-week-old outbred white mice, which were infected in the brain, subcutaneously and intraperitoneally, with a 10% suspension of the sick mice brain of the 2nd and 5th bioisolate passages, and its titer was determined in lg LD_50_/mL.

#### 2.6.2. Sensitivity of Chick Embryos

The study was carried out by infecting 9-day-old chicken embryos with a 0.1 mL virus suspension containing 3 lg PFU on the chorioallantoic envelope (CAE). The results were determined on days 1, 2, and 3 post-infection (pi) by detecting rashes on CAE, as well as by the accumulation of the virus in CAE using the indirect MFA method on the *PEK* cell line and by detecting the virus titer (PFU) on the *Vero-E6* cell line.

#### 2.6.3. Influence of Physical Factors on the Inactivation of a Viral Isolate

The influence of physical factors (temperature 60 °C, 100 °C, ultraviolet irradiation) on the virus inactivation was carried out when exposed to them for 1, 2, 5, and 10 min pi. To verify the results on *PEK* cells, the indirect MFA method was used, and on the *Vero-E6* cell line, the plaque method was used. To assess the effect of ultraviolet radiation on the infectious activity of the pathogen, a small Petri dish with a virus-containing liquid (3 lg PFU) that was 3 mm thick was used. This dish was installed at a distance of 24 cm from a source of short-wave ultraviolet rays with a maximum wavelength of 253.7 nm.

#### 2.6.4. The Study of the Infectious Activity of Viral Particles

After filtration through a 0.22 µm millipore filter (Biofil), the study was carried out using the virological methods described above. 

#### 2.6.5. Electron Microscopic Examination of the Viral Isolate Was Carried out by Method of Negative Staining

Negative contrast electron microscopy was performed on two samples of the isolate. The first sample was the original isolate and the second sample was the isolate after filtration (filter 0.22 nm). The *PEK* cell line was infected with these samples, and the supernatant was collected on days 1–3. Virus particles in the supernatants were visualized by negative staining EM analysis. Supernatants from each sample (1 mL) were centrifuged at 3000× *g* for 20 min at 4 °C to remove cell debris. The clarified supernatants were then centrifuged at 13,000× *g* for 40 min at 4 °C. The pellets were resuspended in 10 μL PBS. Formvar carbon-coated copper grids were floated in droplets of virus suspension for 10 min and stained with 2% phosphotungstic acid for 1 min at room temperature. Subsequently, the grids were examined by transmission electronic microscopy [7]. The samples were viewed using a JEM-100S transmission electron microscope (JEOL, Tokyo, Japan) at an accelerating voltage of 80 kV.

## 3. Results

### 3.1. History of Virus Isolation

In May 2016, a 75-year-old woman with a half-fed *Ixodes persulcatus* tick applied to the laboratory for testing on tick-borne infections. The tick was attached in the territory of the Nadezhdinsky region of Primorsky Krai. The tick was removed five days after feeding on the body of the victim. Due to the fact that tick-borne encephalitis virus (TBEV), the first representative of the *Flaviviridae* family, dominates and is widespread in the Primorsky Territory of the Russian Far East [4], the tick was studied for the detection of the tick-borne encephalitis virus (TBEV) antigen in ELISA, as well as in qPCR for the genetic marker of tick-borne infections (TBEV, *Borrelia burgdorferi* sensu lato, *Anaplasma phagocytophilum*, *Ehrlichia muris/Ehrlichia chaffeensis*). At the time of the study, in the bioassay of a homogenized tick, an antigen of the TBE virus was detected with a low positivity coefficient (K = 1.7); in real-time qPCR, a genetic marker was determined only for the TBE virus and also with a low indicator (Ct = 34), which indicated a low degree of viral load of this pathogen.

A suspension of the homogenized tick was used to infect the brain of outbred white mice of 2 days of age. The brain of one sick suckling mouse with an unclear clinic was taken on the sixth day pi and passaged also by the intracerebral infection of suckling mice of another family. In the first passage, mice began to get sick for 6 to 10 days, and in the second passage, they were sick for 5–7 days, with the maximum activity of clinical manifestations on the fifth day. In the third passage, all infected suckling mice fell ill on the fourth day, and in the fourth and fifth passages, they fell sick on the third day. It was suggested that the studied isolate could be a TBE virus. However, the study of the sick mice brain of all passages did not show, either in ELISA or in qPCR, the antigen or genetic marker of TBEV. It was necessary to understand the reason for the elimination of TBEV from a bioassay of a half-fed tick, in which, as shown above, low rates of infection with this virus were determined.

From the anamnesis, it was found that the patient had previously been vaccinated against TBE. An ELISA study of her blood serum showed the presence of IgG antibodies to the TBE virus in a titer of 1:200. It was suggested that in the process of a tick bloodsucking for 5 days on a vaccinated person, antibodies in such titer could contribute to the neutralization and elimination of the TBE virus in the tick. Thus, in the present case, the tick-borne isolate was obviously not TBEV, but some other unknown virus.

Along with this, the study of the blood serum of this woman by the indirect MFA method for the detection of antibodies to this isolate showed the presence of those with a low titer of 1:8 (Figure 1). No clinical manifestations of the disease were observed in the patient.

### 3.2. Characterization of the Virus In Vivo

In the study of a new unknown pathogen, an important feature is the determination of the sensitivity of various laboratory models to this pathogen. First of all, this refers to the determination of the susceptibility of laboratory animals, in particular white mice, of different age groups with different methods of pathogen inoculation. Figure 2 shows 2-day-old white mice infected with a 10% viral suspension of the second passage in the brain with signs of encephalitis (convulsions, paresis of the limbs).

In the model of white mice of 2 days of age, the titer of the second passage virus was determined, which was 7.2 lg LD_50_/mL with intracerebral (i/c) infection and 5.6 lg LD_50_/mL with subcutaneous (s/c) infection. In the model of white mice of 4 weeks of age, weighing 8–10 g, the virus titer upon i/c infection was 6.0 lg LD_50_/mL, and mice did not become sick with s/c infection (0 lg LD_50_/mL). After intradermal infection with a viral isolate in the paw of the hind limb, swelling of all paws was observed on days 7–10 due to pronounced diffuse edema, and exudation, skin rashes, and necrotic changes were absent. However, when the viral isolate was passaged up to passage 5, its activity increased sharply, the incubation period was reduced to 3 days, the virus titer increased significantly to 10.0 lg LD50/mL, i.e., 10,000 times. The virulence of the virus isolate also notably increased, and it became pathogenic in all methods of inoculation (Figure 3).

### 3.3. Characteristics of the Virus in Experiments In Vitro

To obtain data on the characterization of a new pathogen, it was necessary to study the comparative sensitivity of some cell cultures to it, for which the *PEK*, *Vero*, and *Vero-E6* cell lines were used. Figure 4 shows that a 10% brain suspension of the second passage of suckling mice viral isolate did not form plaques on the *PEK* cell line. On cell lines *Vero* and *Vero-E6*, the plaques were small and dotted, similar to a needle prick. At the same time, on the fifth day pi of the monolayer of cell lines, the virus titer on *Vero* cells was 2 lg PFU lower compared to the virus titer on *Vero-E6* cells.

The infectivity of the viral isolate was shown in MFA starting from the early stages (3 h and 24 h) pi of *PEK*, *Vero*, and *Vero-E6* cells (Figure 5).

This figure shows a bright fluorescence of the antigen in the infected cells of all lines. Despite the fact that the viral isolate did not cause a cytopathic effect and that plaques formed on the *PEK* culture, in the indirect MFA method, the viral antigen formed a thin fluorescent layer on the cell surface as early as 3 h pi, and a more distinct fluorescence of the cytoplasmic membrane of these cells was observed 24 h later. Less-bright fluorescence was obtained from *Vero* cells. *Vero-E6* cells highly sensitive to the virus isolate showed the brightest fluorescence, especially in their cytoplasm. In the infected culture of these cells, after 3 h, one could observe a bright fluorescence in the cytoplasmic membrane, and after 24 h, it was seen not only the membrane, but also the cytoplasm, indicating the active reproduction of the viral isolate in *Vero-E6* cells.

In addition, 5 days pi of *PEK* cells with the second passage viral isolate, it was detected in the supernatant, as well as in the cell sediment during i/c infection of mice 4 weeks old, its titer was low, and it amounted to 2 lg LD_50_/0.03 mL and 1.2 lg LD_50_/0.03 mL, respectively. On this basis, relative to the viral isolate, the *PEK* line can be classified as a permissive cell culture, but with a low level of productive viral infection.

The results obtained above allowed us to continue studying the biological properties of the isolate on *PEK* cells using the indirect method of fluorescent antibodies (MFA). The results were obtained based on the accumulation of the pathogen in the supernatant of the infected culture, collected in the period from 3 h to 9 days pi, with which the cells of the *PEK* line were infected. Figure 6 shows that when the monolayer of the *PEK* line was infected with samples of supernatants, the fluorescence of cells was detected from the first day pi. The active replication of the isolate and its release into the supernatant were confirmed by positive results in MFA, especially from days 5 to 8, when there was a gradual increase in the number of fluorescent cells (47%, 83, 88%, and 95%) of the infected *PEK* line.

### 3.4. The Effect of Physical Factors on the Isolate

To conduct experiments on the action of physical factors, the dynamics of isolate replication in a highly permissive *Vero-E6* cell line 2, 3, 5, and 7 days pi were previously studied (Table 1). Solitary plaques appeared on the monolayer of this cell culture on the second day pi. On day 3, a large number of plaques were observed at a 10^−1^ dilution. On the fifth day, in the cells of the *Vero-E6* line, the isolate actively multiplied to a dilution of 10^−4^. Finally, on the seventh day, we observed its maximum accumulation up to a dilution of 10^−5^ in this permissive medium (Table 1).

Our further studies were aimed at studying the effect of physical factors on the viral isolate. Studies were carried out by inactivating it at a temperature of 60 °C and 100 °C, as well as by ultraviolet irradiation (UVR), when exposed to these factors from 1 to 10 min. Observations were made in parallel on two cell lines, *Vero-E6* and *PEK*, differing in their susceptibility to the studied viral isolate (Table 2 and Table 3).

The reproduction of the virus in *Vero-E6* cells was observed during infection with samples after exposure to a temperature of 60 °C for 1 and 2 min (7 × 10^3^ PFU and 4 × 10^1^ PFU). With further heating of the viral isolate, the ability to form plaques was lost (Table 2). Complete inactivation of the viral isolate was noted after exposure to a temperature of 100 °C. At the same time, the detection of plaques of the viral isolate was observed 1, 3, and even 5 min after exposure to UVI (6 × 10^3^, 11 × 10^1^, and 8 × 10^1^, respectively).

As can be seen in Table 3, heating the viral isolate at 60 °C contributed to a decrease in its activity by 13% at 1 min, by 33% at 2 min, by 37% at 5 min, and by 55% at 10 min of exposure. Heating of the virus isolate at 100 °C, as well as exposure to UVI, contributed to a significant decrease in its activity.

### 3.5. Determining the Size of Viral Particles

A preliminary determination of viral particles size was carried out by filtering the bioisolate through a Millipore filter (0.22 µm). A 10% brain suspension of the sick mice 2 days of age of the fifth passage, taken on the third day pi, was used. At the same time, suckling mice infected in the brain by the resulting filtrate remained healthy for 21 days of observation. Two assumptions were made: mice could not get sick if the virus was larger than 220 nm and it remained on the filter, and they could not get sick if the virus was smaller than 220 nm and it was not pathogenic for outbred suckling mice.

To test this hypothesis, the amount of virus in the bioisolate and in its filtrate was determined. For this purpose, the *Vero-E6* cell line was used (Figure 7). The obtained results indicated that small plaques, the size of a needle prick, were detected in the virus isolate before and after filtration, i.e., the same as in Figure 4. The number of plaques before filtration was 15 × 10^−4^, and after filtration it was 6 × 10^−3^. It was concluded that the number of viral particles after filtration significantly decreased but not completely.

Since the studied isolate was received on the model of outbred white mice, and if the new pathogen had a large size of more than 220 nm, then, first of all, it was necessary to exclude the relationship of the isolate with the mouse ectromelia virus. For this purpose, chicken embryos of 9–10 days of age were infected by introducing a virus-containing material, as well as its filtrate, at a dilution of 10^−1^ by 0.1 mL onto the surface of the allantoic envelope (CAE) of a chicken embryo.

Figure 8A shows the chorioallantoic envelope of chick embryos infected with a viral isolate. Here, on the first day, we did not observe any rashes; throughout the third to seventh days, typical rash characteristic of the smallpox virus—white pockmarks (plaques)—were detected. During these studies, we did not observe hemorrhagic pockmarks. The chorioallantoic envelope of chicken embryos contaminated with filtrate remained without rashes (Figure 8B). Figure 8 was put into the appendix file.

To prove the specific effect of the virus on chicken embryos, a monolayer of the *PEK* line was infected with a suspension of samples of the chorioallantoic envelope; at three days pi, bright fluorescence was detected in *PEK* cells in IMPA, indicating the presence of the antigen in samples of both the initial isolate and its filtrate. In addition, the virus titer was determined on the *Vero-E6* cell line. The virus isolate at 1 day pi was not detected, and after 3 and 7 days, its titer was 3 lg and 5 lg PFU. In samples of CAE chicken embryos contaminated with filtrate, the virus was not verified during all periods of observation.

Based on the obtained results of a comprehensive study, it was suggested that the isolate from the tick *I. persulcatus* contains at least two viruses. To test this hypothesis, it was necessary to receive the direct evidence using the method of electron microscopy.

### 3.6. Electron Microscopy of Viral Co-Isolate

To confirm the above assumption, electron microscopic studies of the co-isolate were carried out using the negative staining method. In the cells of the *PEK* line infected with the co-isolate, on the third day pi, two types of viral particles were verified, which differed from each other. Two viruses were detected in the sample: one that was large and smallpox-like, up to 500 nm, and the other much smaller, from 60 to 100 nm (Figure 9). In addition, conglomerates consisting of several viral particles could be observed here, on the surface of which adherent particles of small sizes were sometimes observed.

For better visualization of both co-infection viruses, the samples were prepared from the supernatant of the *PEK* cell line, taken after 1 day pi, that is in the early period of viral replication, as shown in the IMFA in Figure 5 and Figure 6.

Figure 10A shows a significant predominance of large particles up to 1000 nm, which, in our opinion, consists of 2–3 confluent smallpox-like virus particles. There was a fragmented rim along the periphery of such an electron-dense conglomerate. Figure 10B–D shows that the conglomerates consist of merged, apparently pox-viral particles, and the rim had adhered particles of a small virus, up to 100 nm in size. In our opinion, such a morphological picture demonstrated the phenomenon of the cocultivation of two viruses. At the same time, free-lying particles of a small virus, up to 100 nm in size, were also found.

Thus, based on biological characterization results and using electron microscopy, it was shown that the viral isolate presented as a co-infection of two pathogens: a representative of the smallpox virus family and a pathogen unknown to us.

## 4. Discussion

Currently, researchers are paying close attention to the study of arthropod infection with known and unknown pathogens that represent a danger not only to the human population, but also to wild and domestic animals. Modern publications on the verification of pathogens from different members of natural foci of infection, as a rule, are based on the results of the genetic markers detected in qPCR and other molecular biological studies [2,3,21]. Widely conducted studies on the prevalence of tick-borne pathogens in various focal areas have significantly enriched our understanding of the biodiversity of the ixodid tick virome [22]. In addition, there are often cases of mixed infection of ticks simultaneously with various pathogens [23]. Probably, in such cases, favorable conditions arise for the equilibrium co-cultivation of several pathogens in the same biological object [24].

In this work, we also had to solve the issues of the interaction of several pathogens in the microbiome of the tick *I. persulcatus*. In the process of studying the bioisolate, we were able to verify three different viruses. Initially, in the test tick after 5 days of bloodsucking on a human, infection with tick-borne encephalitis virus (TBEV) was established at low rates in ELISA and qPCR. As it turned out, specific post-vaccination antibodies to TBEV were present in the blood of the examined patient, which probably could contribute to the neutralization and elimination of this pathogen in the tick during bloodsucking. This outcome could be supported by the results of our long-term studies in vitro, ex vivo, and in vivo, aimed at studying the effectiveness of specific protection against the TBE virus [25]. Using a clear biological model—the blood of vaccinated individuals with different titers of antibodies to TBEV: 1:100; 1:200; 1:400; 1:800; 1:1600; 1:3200—we showed that in samples with specific antibodies in titers of more than 1:400, the neutralization of the virus occurred quickly (after 24 h). Under the action of antibodies in titers of 1:100 and 1:200, the elimination of the pathogen also occurred, but at a later date, on days 3–4 pi of these samples with the TBE virus [26]. This allowed us to assume that specific antibodies circulating in the patient’s blood in a titer of 1:200 during tick bloodsucking could affect the suppression of the viability of the TBE virus, as a result of which it was not isolated during virological studies, and the patient did not get sick with TBE.

Such a classical concept of antibody-dependent elimination of the TBE virus is quite understandable. However, there is an assumption that, in the case of co-infection of the TBE virus and a new virus, the possibility of TBEV elimination can be assumed if the new virus acts as a suppressor during its replication in the tick body. In this regard, it should be noted that for almost two decades we have been observing a gradual decrease in the infection rates of TBEV ticks collected from vegetation in natural foci, as well as ticks attached to people. This downward trend in the TBE virus infection of ticks was reflected in TBE incidence rates in the south of the Far East, where isolated cases of this infection have been recorded in the population in the last decade [27].

In the case under study, the previously unknown pathogen of the coinfection we identified in the tick microbiome received the opportunity for its replication and the appearance of specific antibodies in the patient, although its titer was low (Figure 1). Earlier, for 2 years, we noted in one patient the clinical and diagnostic features of a mixed infection caused by TBEV, *Borrelia*, and SARS-CoV2 [28]. Using the indicators of antigen detection in ELISA, genetic markers in qPCR, and antibodies, as well as the degree of their avidity to these pathogens, we found that in the case of detection of three infections simultaneously, the bacterial pathogen *B. burgdorferi dominates* in the human body, causing a long-term chronic course of the disease [27]. Such dominance of *Borrelia* in the body of a tick infected with the TBEV and *B. burgdorferi* was previously proven in the experiment [29].

This means that, in cases of the co-infection of different pathogens, any pathogen can dominate. In the described case, when studying the biological characteristics of the viral isolate, we observed an increase in the activity of the mouse pox virus. The isolation of a new virus in ticks can be considered an accident because, as it was found, this virus is not pathogenic for white mice. However, this chance was predetermined by the fact that the isolation of the virus was performed on a model of outbred mice of 2 days of age, which received the ectromelia virus vertically from the mother—the carrier of this chronic infection. We failed to isolate the TBE virus, but an unknown pathogen was identified, which, in our opinion, was a trigger for the activation of a latent chronic infection in sucklings of outbred white mice. A similar case of the coinfection of two viruses (murine ectromelia virus and lactate dehydrogenase-elevating virus) in ticks was described in 2021 [20]. However, the authors failed to pass the second LDV virus.

Using various virological research methods, we focused our efforts on identifying the features of the biological characteristics of the isolated viral co-isolate. In favor of the presence of the ectromelia virus in it, the following manifestations were testified: paw edema during the intradermal infection of adult mice, characteristic rashes on the CAE of infected chicken embryos, typical plaques in the form of an injection on the infected *Vero-E6* culture, instability to warming, and resistance to ultraviolet irradiation. However, an electron microscopic examination showed that in addition to the smallpox-like virus, the co-isolate contained another virus that notably differed from the smallpox virus.

Although the new virus we identified turned out to be non-pathogenic for white mice of all age groups and chicken embryos, it did not cause cytopathogenic effects and plaque formation on PEK cells, but it successfully multiplied in *PEK*, *Vero*, and *Vero-E6* cell lines. Its presence in the viral co-isolate is shown in the electron microscopy images.

Based on the obtained results, this viral co-isolate can be characterized as a stable form of the co-cultivation of murine ectromelia virus and a new tick-borne virus, named *“Kiparis-144*”. Moreover, the dominant mouse ectromelia virus served as an indicator for the verification of the second virus, *Kiparis-144*. This is probably why, without such an indicator and without a permissive model for virus isolation, researchers [2,3,20] failed to isolate new pathogens in molecular genetic studies of the ixodid tick virome.

The interaction of these two viruses contributed to a significant increase in the virulence of the viral co-isolate from passages 2 to 5 in outbred white mice with all methods of inoculation. The incubation period in suckling mice infected in the brain was reduced from 5–7 days to 2.5–3 days. The virus titer after the intracerebral infection of 3-week-old mice increased by 4 lg LD_50_, i.e., 10,000 times. This indicated that when two viruses were co-cultivated in warm-blooded animals, the new virus acted as an amplifier or “accelerator” of mouse pox virus replication. Their combined interaction was clearly shown by us using electron microscopy (Figure 9).

In epidemiological and epizootological terms, this property of the new virus is a potentially dangerous and naturally causes concern. Moreover, in recent years, natural populations of various representatives of the smallpox virus have become more active in the world [30,31,32]. We admit the possibility of expanding the ecological and virological competencies of ticks containing pathogens such as the *Kiparis-144* virus. When bloodsucking ticks with such viruses are found on cows, monkeys, donkeys, and other animals in cases of chronically occurring smallpox virus infection, its activation can occur with the involvement of a human in this process.

## 5. Conclusions

Thus, the obtained results allow us to conclude that it is necessary to conduct further scientific research to study the molecular genetic characteristics of not only the ectromelia virus, but also the new *Kiparis-144* virus detected in the half-fed tick *I. persulcatus* in the south of the Far East, as well as to determine the degree of its pathogenicity for humans and animals.

## Figures and Tables

**Figure 1 microorganisms-11-01791-f001:**
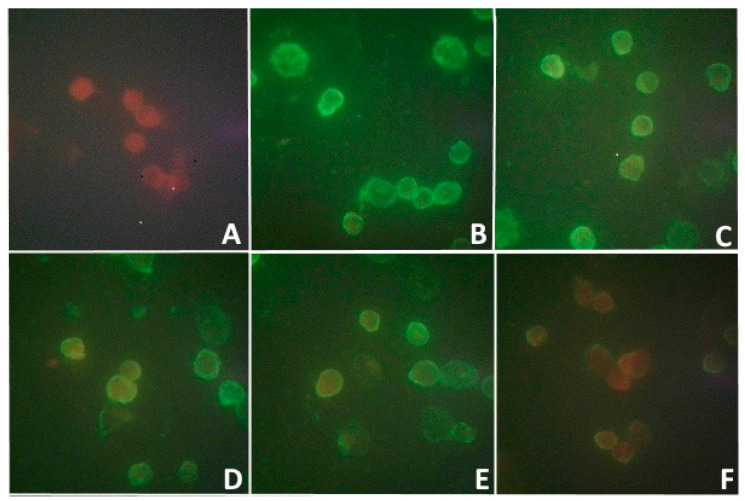
Verification of antibodies to a new isolate in the blood serum of a patient who removed a tick from himself on the 5th day of bloodsucking. Indirect MFA method on the *PEK* cell line model. Note: (**A**) control of uninfected cells; (**B**) original serum; (**C**) serum diluted 1:2; (**D**) serum diluted 1:4; (**E**) serum diluted 1:8; (**F**) serum diluted 1:16. ×650.

**Figure 2 microorganisms-11-01791-f002:**
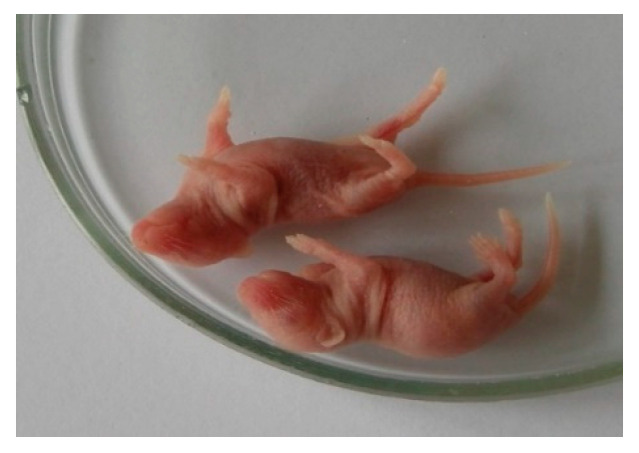
Clinical manifestations in mice of 2 days of age on the 5th day after intracerebral infection with the isolate (2nd passage).

**Figure 3 microorganisms-11-01791-f003:**
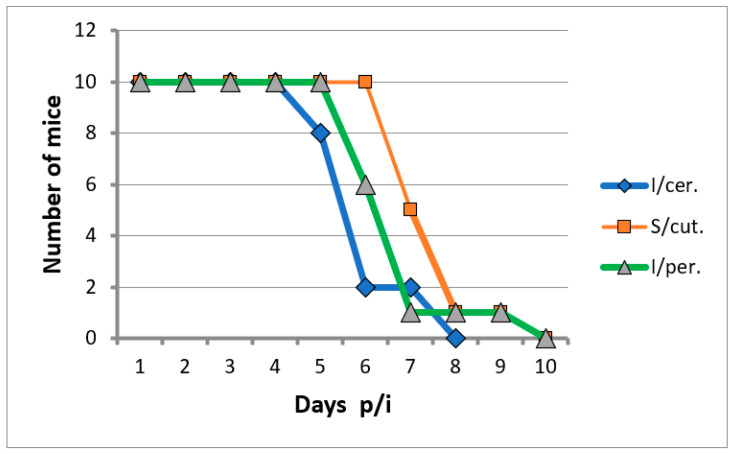
The death of white mice during intracerebral (i/cer), subcutaneous (s/cut), and intraperitoneal (i/per) infection with the viral isolate of the 5th passage (dilution 10^−1^, which amounted to 3 lg PFU).

**Figure 4 microorganisms-11-01791-f004:**
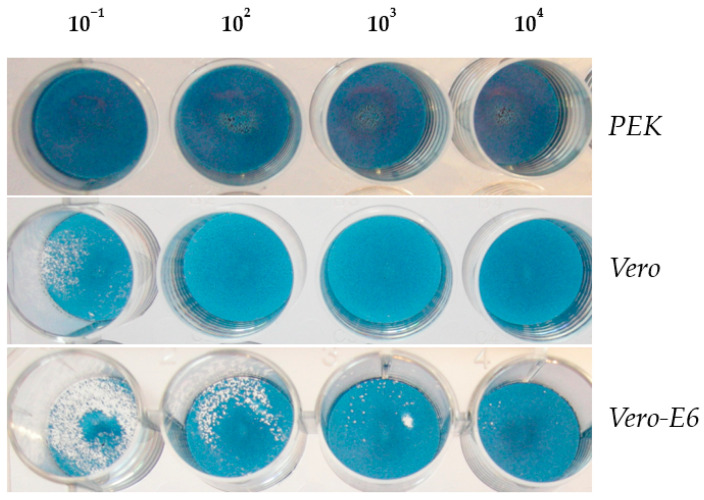
Plaque-forming ability of the bioisolate on cell lines *PEK*, *Vero*, and *Vero-E6* (on the 5th day after infection).

**Figure 5 microorganisms-11-01791-f005:**
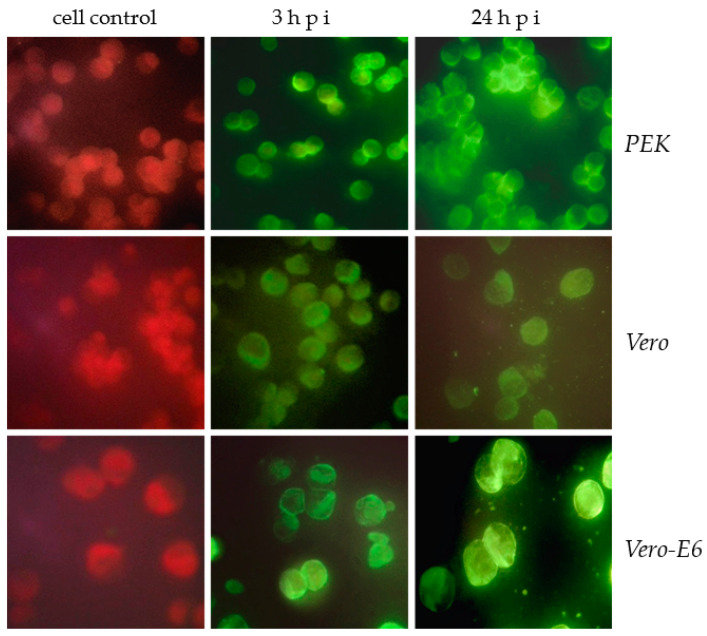
Specific antigen fluorescence in *PEK*, *Vero*, and *Vero-E6* cells infected with a viral isolate 3 and 24 h pi. MFA method: *PEK* cells; *Vero* cells; *Vero-E6* cells. ×650.

**Figure 6 microorganisms-11-01791-f006:**
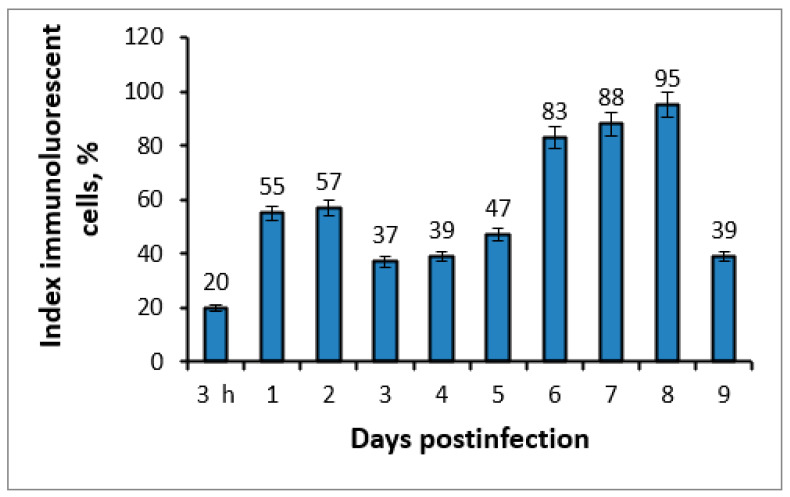
Accumulation of the viral isolate in the supernatant in the period from 3 h to 9 days pi of the *PEK* cell line. The visualization of the antigen on cells using the method of fluorescent antibodies shows the average level of antigen-positive cells (%).

**Figure 7 microorganisms-11-01791-f007:**
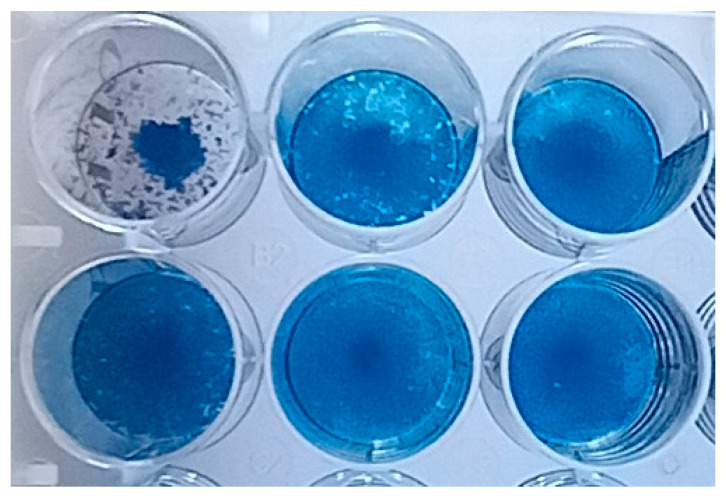
Verification of the viral isolate in the *Vero-E6* cell culture 5 days post-infection (pi). Titer of virus isolate before filtration (1st row) and after filtration (2nd row).

**Figure 8 microorganisms-11-01791-f008:**
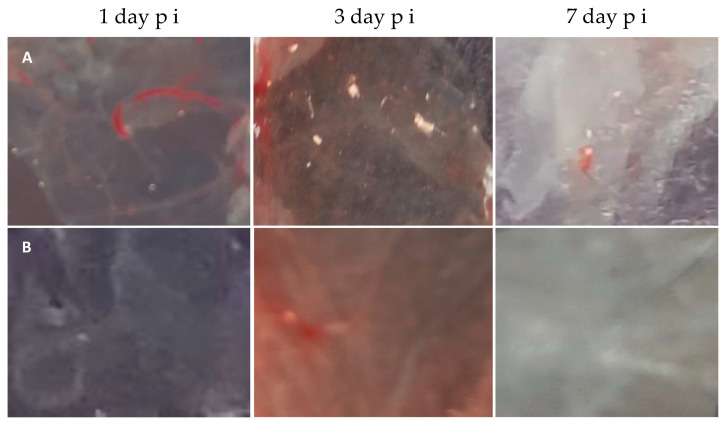
Chorion-allantoic envelope of chicken embryos 9–10 days old, infected with a suspension of the original viral isolate (**A**) and its filtrate (**B**) on days 1, 3 and 7–9 postinfetion (pi).

**Figure 9 microorganisms-11-01791-f009:**
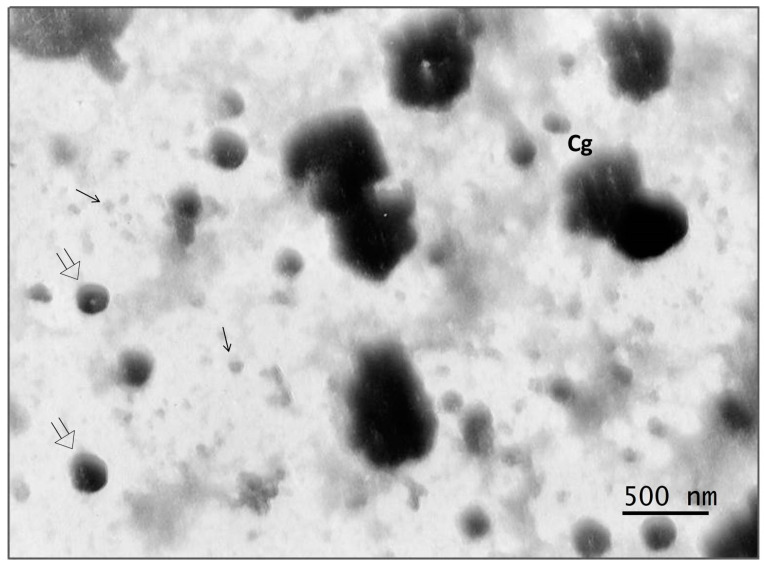
The virus co-isolate from the tick *I. persulcatus* that half-fed on a human. Two viruses of different sizes are visible: one large and smallpox-like, up to 500 nm (double arrow); the other is much smaller, from 60 to 100 nm (arrow). Electron microscopy with negative contrast.

**Figure 10 microorganisms-11-01791-f010:**
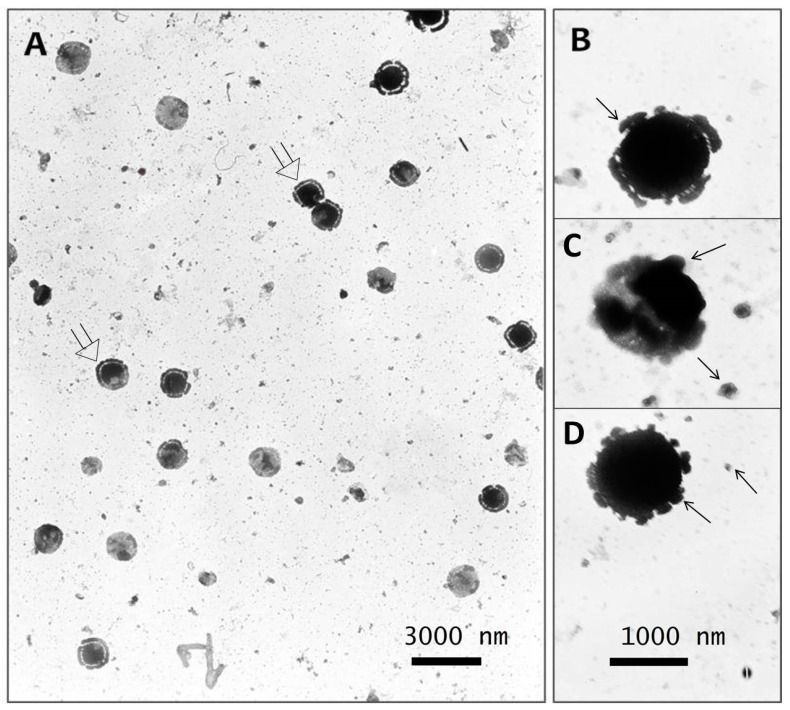
Virus co-isolate from the tick *I. persulcatus* that half-fed on a human. (**A**) A significant predominance of large viral particles up to 500 nm in size; (**B**–**D**) electron-dense conglomerates, which consist of fused pox-like viral particles with a rim around the periphery, representing adherent particles of a small virus, up to 100 nm in size.

**Table 1 microorganisms-11-01791-t001:** Dynamics of the plaque-forming ability of the isolate at its tenfold dilutions from 10^−1^ to 10^−5^ in *Vero-E6* cells on the 2nd, 3rd, 5th, and 7th days pi.

Observation Day	Number of Plaques, PFU				
	10^−1^	10^−2^	10^−3^	10^−4^	10^−5^
2	single	0	0	0	0
3	multiple	0	0	0	0
5	confluent	multiple	26	13	0
7	confluent	confluent	multiple	15–17	1–3

**Table 2 microorganisms-11-01791-t002:** Influence of physical factors (temperature and UVR) on the viral isolate during exposure from 1 to 10 min. Determination of the plaque-forming ability of the virus in the permissive cell line *Vero-E6*.

№	Physical Factors	Exposure Time			
		1 min	2 min	5 min	10 min
1	Heat inactivation of the virus, 60 °C (lg PFU)	7 × 10^3^	4 × 10^1^	0	0
2	Heat inactivation of the virus, 100 °C (lg PFU)	0	0	0	0
3	UVI virus inactivation (lg PFU)	6 × 10^3^	11 × 10^1^	8 × 10^1^	0
4	Virus control (lg PFU)	6 × 10^4^			

**Table 3 microorganisms-11-01791-t003:** Influence of physical factors (temperature and UVI) on the viral isolate during exposure from 1 to 10 min. Visualization of the pathogen by detecting the fluorescence of *PEK* cells in IMFA.

№	Physical Impact	Exposure Time			
		1 min	2 min	5 min	10 min
1	Heat inactivation of the virus, 60 °C (%)	63	43	39	21
2	Heat inactivation of the virus, 100 °C(%)	36	23	17	4
3	UVI virus inactivation (%)	30	25	17	7
4	Virus control	76% of cells with antigen (IMPA+)			

## Data Availability

This article does not contain any studies involving humans as primary objects of research.

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
