# Peer review of "Study of Viral Coinfection of the Ixodes persulcatus Ticks Feeding on Humans in a Natural Focus of the South of the Far East"

_microorganisms, 2023, doi:10.3390/microorganisms11071791_

Round 1

Reviewer 1 Report

The manuscript entitled as “Study of viral coinfection of the Ixodes persulcatus tickets 2 feeding on humans in a natural foci of the south of the Far East” is a good work. However, the following suggestions should be noted.

Line 2: “ixodes” should be changed as “Ixodes”.

Line 45: “Cervus nippon” should be italic.

Line 70: “Ixodes persulcatus” should be changed as “I. persulcatus” (italic).

Lines 83-85: The sentence“Borrelia burgdorferi sensu lato, Anaplasma phagocytophilum, Ehrlichia muris/Ehrlichia chaffeensis by real-time polymerase chain reaction (RT-PCR) using the kit "AmpliSense TBEV, B. burgdorferi s.l., A. phagocytophilum, E. chaffeensis / E. muris” includes some pathogen nons. The first appearance should be italic with full name. The second appearance should be also italic with the genus name being abbrivated.

Lines 152-153: The sentence “Borrelia burgdorferi sensu lato, Anaplasma phagocytophilum, Ehrlichia muris/Ehrlichia chaffeensis” should be changed as “B. burgdorferi s.l, A. phagocytophilum, E. muris/E. chaffeensis” (the pathogen name should be italic).

Line 283: 3 should be uperscripted.

Figure 8 should be put into appendix file.

The manuscript entitled as “Study of viral coinfection of the Ixodes persulcatus tickets 2 feeding on humans in a natural foci of the south of the Far East” is a good work. However, the following suggestions should be noted.

Line 2: “ixodes” should be changed as “Ixodes”.

Line 45: “Cervus nippon” should be italic.

Line 70: “Ixodes persulcatus” should be changed as “I. persulcatus” (italic).

Lines 83-85: The sentence“Borrelia burgdorferi sensu lato, Anaplasma phagocytophilum, Ehrlichia muris/Ehrlichia chaffeensis by real-time polymerase chain reaction (RT-PCR) using the kit "AmpliSense TBEV, B. burgdorferi s.l., A. phagocytophilum, E. chaffeensis / E. muris” includes some pathogen nons. The first appearance should be italic with full name. The second appearance should be also italic with the genus name being abbrivated.

Lines 152-153: The sentence “Borrelia burgdorferi sensu lato, Anaplasma phagocytophilum, Ehrlichia muris/Ehrlichia chaffeensis” should be changed as “B. burgdorferi s.l, A. phagocytophilum, E. muris/E. chaffeensis” (the pathogen name should be italic).

Line 283: 3 should be uperscripted.

Figure 8 should be put into appendix file.

Author Response

Responses to Reviewer 1

Dear reviewer, thank you very much for evaluating our work, your comments have been corrected.

 Line 2:  changed “Ixodes

 Line 45: changed “Cervus nippon”

Line 70: “Ixodes persulcatus” changed “I. persulcatus

Lines 83-85: changed italic

Lines 152-153: changed italic

Line 283: changed

Figure 8 put into appendix file.

Reviewer 2 Report

This is an interesting manuscript that provides new information on viral infections in ticks. The article is generally well written, but some passages are confusing and should be rewritten.

Title

ticks instead of “tickets”

focus instead of “foci”

Abstract

L22-23: “which poses a serious epidemiological danger”- this statement is rather speculative and should be moderated

Introduction

L 44: Earlier, in 1971, the Kansas virus (KHAV) was isolated from ticks….

Materials and Methods

Authors should consider including most part of information contained in item “3.1 History of virus isolation” in the Material and Methods section, instead of in the Results section.

L70: “An unknown pathogen” instead of “An unknown for us pathogen”

L81: qPCR instead of RT-PCR. Please correct it throughout the manuscript.

L91: “infected” instead of “contaminated”.

L117: “Influence of physical factors on the inactivation of a viral isolate” instead of “Influence of physical factors on the ell line. inactivation of a viral isolate”

L118: Please exclude “Influence of physical factors on the inactivation of a viral isolate”

L119: “60ºC” instead of “60º” and “100ºC” instead of “100º”. Please correct throughout the manuscript.

Results:

Item 3.1 should be in the Material and Methods section

L146-147: “The tick was sucked on the territory of the Nadezhdinsky region of Primorsky Krai.” It is not clear what the authors mean by this sentence, please rewrite it.

L148: “patient” instead of “victim”.

L171-173: “It has been suggested that in the process of 171 bloodsucking the tick for 5 days on a human with antibodies to the TBE virus in a titer of 172 1:200, neutralization and elimination of the TBE virus occurred in the fed tick”. This sentence is very confusing, please consider rewriting it.

Figure 3: abbreviations should be avoided in figure captions.

Author Response

Dear reviewer, thank you very much for evaluating our work, your comments have been corrected.

Title

ticks instead of “tickets” - changed

focus instead of “foci” - changed

Abstract

L22-23: changed - an epidemiological danger

L 44: changed

In the Results section 3.1 “History of virus isolation” - changed

L70: changed

L81: changed

L91: changed

L117: changed

L118: changed

L119: changed

Results: Item 3.1; L146-147; L148 - deleted

L171-173: It has been suggested that during 5 days of tick bloodsucking on a vaccinated person, antibodies in a titer of 1:200 could contribute to the neutralization and elimination of the TBE virus. Thus, in the present case, the tick-borne isolate is obviously not TBEV, but some other unknown virus.

Corrected paragraph:

From the anamnesis it was found out that the patient had previously been vaccinated against TBE. An ELISA study of her blood serum showed the presence of IgG antibodies to the TBE virus in a titer of 1:200. It was suggested that in the process of bloodsucking a tick for 5 days on a vaccinated person, antibodies in such titer could contribute to the neutralization and elimination of the TBE virus in the tick. Thus, in the present case, the tick-borne isolate is obviously not TBEV, but some other unknown virus.

Figure 3: corrected - The death of white mice during intracerebral (i/cer), subcutaneous (s/cut) and intraperitoneal (i/per) infection with the viral isolate of the 5th passage (dilution 10-1, which amounted to 3 lg PFU).

Dear reviewer, thank you very much for evaluating our work, your comments have been corrected.

Title

ticks instead of “tickets” - changed

focus instead of “foci” - changed

Abstract

L22-23: changed - an epidemiological danger

L 44: changed

In the Results section 3.1 “History of virus isolation” - changed

L70: changed

L81: changed

L91: changed

L117: changed

L118: changed

L119: changed

Results: Item 3.1; L146-147; L148 - deleted

L171-173: It has been suggested that during 5 days of tick bloodsucking on a vaccinated person, antibodies in a titer of 1:200 could contribute to the neutralization and elimination of the TBE virus. Thus, in the present case, the tick-borne isolate is obviously not TBEV, but some other unknown virus.

Corrected paragraph:

From the anamnesis it was found out that the patient had previously been vaccinated against TBE. An ELISA study of her blood serum showed the presence of IgG antibodies to the TBE virus in a titer of 1:200. It was suggested that in the process of bloodsucking a tick for 5 days on a vaccinated person, antibodies in such titer could contribute to the neutralization and elimination of the TBE virus in the tick. Thus, in the present case, the tick-borne isolate is obviously not TBEV, but some other unknown virus.

Figure 3: corrected - The death of white mice during intracerebral (i/cer), subcutaneous (s/cut) and intraperitoneal (i/per) infection with the viral isolate of the 5th passage (dilution 10-1, which amounted to 3 lg PFU).

Dear reviewer, thank you very much for evaluating our work, your comments have been corrected.

Title

ticks instead of “tickets” - changed

focus instead of “foci” - changed

Abstract

L22-23: changed - an epidemiological danger

L 44: changed

In the Results section 3.1 “History of virus isolation” - changed

L70: changed

L81: changed

L91: changed

L117: changed

L118: changed

L119: changed

Results: Item 3.1; L146-147; L148 - deleted

L171-173: It has been suggested that during 5 days of tick bloodsucking on a vaccinated person, antibodies in a titer of 1:200 could contribute to the neutralization and elimination of the TBE virus. Thus, in the present case, the tick-borne isolate is obviously not TBEV, but some other unknown virus.

Corrected paragraph:

From the anamnesis it was found out that the patient had previously been vaccinated against TBE. An ELISA study of her blood serum showed the presence of IgG antibodies to the TBE virus in a titer of 1:200. It was suggested that in the process of bloodsucking a tick for 5 days on a vaccinated person, antibodies in such titer could contribute to the neutralization and elimination of the TBE virus in the tick. Thus, in the present case, the tick-borne isolate is obviously not TBEV, but some other unknown virus.

Figure 3: corrected - The death of white mice during intracerebral (i/cer), subcutaneous (s/cut) and intraperitoneal (i/per) infection with the viral isolate of the 5th passage (dilution 10-1, which amounted to 3 lg PFU).

Dear reviewer, thank you very much for evaluating our work, your comments have been corrected.

Title

ticks instead of “tickets” - changed

focus instead of “foci” - changed

Abstract

L22-23: changed - an epidemiological danger

L 44: changed

In the Results section 3.1 “History of virus isolation” - changed

L70: changed

L81: changed

L91: changed

L117: changed

L118: changed

L119: changed

Results: Item 3.1; L146-147; L148 - deleted

L171-173: It has been suggested that during 5 days of tick bloodsucking on a vaccinated person, antibodies in a titer of 1:200 could contribute to the neutralization and elimination of the TBE virus. Thus, in the present case, the tick-borne isolate is obviously not TBEV, but some other unknown virus.

Corrected paragraph:

From the anamnesis it was found out that the patient had previously been vaccinated against TBE. An ELISA study of her blood serum showed the presence of IgG antibodies to the TBE virus in a titer of 1:200. It was suggested that in the process of bloodsucking a tick for 5 days on a vaccinated person, antibodies in such titer could contribute to the neutralization and elimination of the TBE virus in the tick. Thus, in the present case, the tick-borne isolate is obviously not TBEV, but some other unknown virus.

Figure 3: corrected - The death of white mice during intracerebral (i/cer), subcutaneous (s/cut) and intraperitoneal (i/per) infection with the viral isolate of the 5th passage (dilution 10-1, which amounted to 3 lg PFU).

Dear reviewer, thank you very much for evaluating our work, your comments have been corrected.

Title

ticks instead of “tickets” - changed

focus instead of “foci” - changed

Abstract

L22-23: changed - an epidemiological danger

L 44: changed

In the Results section 3.1 “History of virus isolation” - changed

L70: changed

L81: changed

L91: changed

L117: changed

L118: changed

L119: changed

Results: Item 3.1; L146-147; L148 - deleted

L171-173: It has been suggested that during 5 days of tick bloodsucking on a vaccinated person, antibodies in a titer of 1:200 could contribute to the neutralization and elimination of the TBE virus. Thus, in the present case, the tick-borne isolate is obviously not TBEV, but some other unknown virus.

Corrected paragraph:

From the anamnesis it was found out that the patient had previously been vaccinated against TBE. An ELISA study of her blood serum showed the presence of IgG antibodies to the TBE virus in a titer of 1:200. It was suggested that in the process of bloodsucking a tick for 5 days on a vaccinated person, antibodies in such titer could contribute to the neutralization and elimination of the TBE virus in the tick. Thus, in the present case, the tick-borne isolate is obviously not TBEV, but some other unknown virus.

Figure 3: corrected - The death of white mice during intracerebral (i/cer), subcutaneous (s/cut) and intraperitoneal (i/per) infection with the viral isolate of the 5th passage (dilution 10-1, which amounted to 3 lg PFU).

Reviewer 3 Report

The authors attempted to study the phenomenon of pathogens co-infection detected in the half fed on humans I. persulcatus tick in the south of the Far East. The authors were also able  to verify three different viruses. This is a novel and commendable work. However, there are some concepts that need clarity:

1. The patient was vaccinated against TBEV and the authors attribute the elimination of TBEV in the tick to this. How sure are they that the other viruses are not suppressing TBEV?

2. The methods used are not adequately described

The quality of English language is poor but intelligible. The authors need to do a thorough English language editing

Author Response

Dear reviewer, thank you very much for evaluating our work, your comments have been corrected.

Paragraph L171-173 has been corrected, where indicate the presence of TBEV antibodies only. For other viruses, more research is needed.

The chapter "Materials and Methods" describes the main stages of the well-known classical methods of virological research. Reference is made to the electron microscopic method of negative contrasting [7].